# Polyurethane/Liquid Crystal Microfibers with pDNA Polyplex Loadings for the Optimal Release and Promotion of HUVEC Proliferation

**DOI:** 10.3390/pharmaceutics14112489

**Published:** 2022-11-17

**Authors:** Chaowen Zhang, Lu Lu, Ruoran Ouyang, Changren Zhou

**Affiliations:** Biomaterial Research Laboratory, Department of Material Science and Engineering, College of Chemistry and Materials, Jinan University, Guangzhou 510632, China

**Keywords:** PU, liquid crystal, microfibers, plasmids, sustained release, interaction with cells

## Abstract

Fiber structures with connected pores resemble the natural extracellular matrix (ECM) in tissues, and show high potential for promoting the formation of natural functional tissue. The geometry of composite fibers produced by electrospinning is similar to that of the living-tissue ECM, in terms of structural complexity. The introduction of liquid crystals does not affect the morphology of fibers. The composite mat shows better hydrophilicity, with higher content of liquid crystal. At the same time, the higher the content of liquid crystal, the lower the modulus and tensile strength, and the higher the breaking energy and the elongation at break. Additionally, the factors affecting fibers are also investigated in this study. The addition of liquid crystals to the fibers’ matrix can slow down the release of pDNA, which is the most common vehicle for genetic engineering, and the encapsulation of pDNA polymer into the fiber matrix can maintain biological activity. The continued release of the pDNA complex was achieved in this study through liquid crystals, and the effective release is controllable. In addition, the integration of liquid crystals into fibers with pDNA polymers can cause a faster transfection rate and promote HUVEC (Human Umbilical Vein Endothelial Cells) growth. It is possible to combine electrospun fibers containing LC (liquid crystal) with pDNA condensation technology to achieve the goal of a sustained release. The production of inductable tissue-building equipment can manipulate the required signals at an effective level in the local tissue microenvironment.

## 1. Introduction

Polyurethanes (PU) have attracted much attention, and are considered a kind of multifunctional polymer that consists mainly of urethane/urea groups [1,2,3]. Otto Bayer first synthesized PU during the Second World War. Since then, PU has been studied by many research groups worldwide. Currently, PU is widely used in many different applications, such as coatings, adhesives and biomaterials [4,5,6]. Due to its adjustable composition, polyurethane has a wide range of mechanical properties (from rubber to plastic) with good thermal stability and functions, and has become a common choice for the preparation of drug delivery devices. The stimulus-responsive structures in polyurethane systems have attracted considerable interest, and can be used for specific targeted delivery of many different types of drug loads [7,8,9,10]. However, biocompatibility cannot easily keep up with the growing demand. Thus, in this study, a kind of cholesteric liquid crystal was synthesized and introduced into PU to enhance the biocompatibility and control the release of pDNA to promote HUVEC (human umbilical vein endothelial cell) proliferation.

Liquid crystals (LCs), another important class of soft matter systems, are formed by weakly interacting anisotropic building blocks, such as molecules or micelles, with anisometric rod-like or disc-like shapes [11,12,13]. In the biological field, the liquid crystal state is a common state. Many tissues in the human body are composed of liquid crystal structures, such as the brain, muscles, blood, ovaries and tendons. In addition to traditional engineering applications, liquid crystal elastomers (LCEs) have been proposed for biomedical applications. These applications include porous tissue-engineered mats, drug delivery vehicles and vascular implants. Most of the LCEs studied are in the crystalline state at body temperature, not the liquid crystal state, which differs from the human body structure of the liquid crystal. A body-temperature liquid crystal is synthesized by us and introduced into material to enhance its biocompatibility.

In recent years, the application of electrospun microfiber components in the biomedical field has attracted much attention [14,15,16,17,18,19,20,21,22]. Cell electrospinning has become a novel tool for functionalizing fibres, scaffolds, mats with living cells and other advanced materials for regenerative biology and medicine [23,24]. Electrospinning is a special fiber-making process in which a polymer solution or melt is sprayed in a strong electric field. Under an electric field, the droplets at the tip of the needle change from spherical to tapered (Taylor’s cone), and extend from the tip to the filament. In this way, polymer filaments with micrometers in diameter can be produced. The main reasons are the large controllable surfaces and openings, porous geometry, fiber size and fiber orientation. The connected porous geometry mimics the topology of the extracellular matrix (ECM) [25,26,27]; this is very important for transporting oxygen and nutrients to cells. The relaxation and combination of fibers promote the movement and distribution of cells across the fibers. Due to the flexibility and high quality of the surface of the substrate materials, electrospun microfibers have become a sustainable medium for antibiotics, antitumor drugs and proteins [28,29,30,31]. The absorption curve of the product can be adjusted according to the fiber morphology, porosity and composition. The absorption capacity of the medicines is combined with tissue-engineered mats to continuously release the drug and maintain a high concentration of bioactive substances. They can not only support cell proliferation and metastasis, but also improve the secretion and function of the ECM. Thus, the electrospinning technique is used in this study to enhance the biocompatibility of the materials and the contact of pDNA with the cells.

Due to the easy deactivation of growth factors, encapsulation in electrospun fibers is an alternative method, which can prolong the shelf life and protein expression for a duration similar to that of the release of fiber growth factors. The incorporation of pDNA into electrospun microfibers has been studied according to previous research [32,33,34,35]. Depending on the type of material, pDNA exists on the surface of fibers, not surrounded by fibers. Compared to a positive control group, the transfection rate of the pDNA released from electric fibers decreases significantly. Unlike viruses, virus protein channels are sealed by narrow DNA molecules. Internally, for the insertion of particle pDNA into electrospun fibers and to avoid being attacked by released enzymes, the electrospun fiber equipment is a better choice than the charged fibers for the preparation of proteins with a core shear structure in the initial research. We delay the initial solution to maintain biological activity.

Although liquid crystals own good fluidity in the melting state or the solution state, the arrangement of molecules in the liquid crystals still shows a certain degree of order, thus showing anisotropy in their physical properties [36,37,38]. Liquid crystalline materials show different colors, structures and appearances in polarized light depending on their orientation. In the biological field, the liquid crystal state is the more common state. Many tissues in the human body are composed of lyotropic liquid crystal structures, such as the brain, muscles, blood, ovaries and tendons [39]. The liquid crystal state is a guarantee of the organic combination and information communication between cells, tissues and organs in the living body, which can help to maintain the balance of the body’s nutrients and the smooth transmission of biological information. There are many types of liquid crystals, such as nematic and near-crystalline liquid crystals, among which the structure of cholesteric (Ch*-LC) crystals is more complex [40,41]. Many natural polymers can spontaneously form a chemical helix structure, much like the self-helices of cholesteric liquid crystal molecules. Cholesteric liquid crystals, due to their proximity to natural polymers, usually have better biocompatibility than nematic and near-crystalline liquid crystals, and can interact with some biomolecules.

In this study, we introduced liquid crystals into PU microfibers as a vehicle for the sustained release of pDNA, for the purpose of enhancing the biocompatibility of the material and providing a better slow release effect. Additionally, the interaction of the composite fibers with HUVEC was studied.

## 2. Materials and Methods

### 2.1. Materials and Cells

PU (3575A) was purchased from Lubrizol Company without further purification (commercially medical grade polyurethane, density 1.15 g/cm^3^, bending modulus 4.27 MPa, tensile strength 36.5 MPa, shore A 73). Cholesteric liquid crystal was synthesized according to our previous study [42,43]. The esterification reaction of polydiols with cholesteric acyl chloride was briefly carried out, and then the hydroxyl end groups of the above compounds were esterified with acrylamide chloride to form cholesteric liquid crystals with double bonds. Finally, a liquid crystal polymer was prepared by hydrosilylation of siloxane with terminal double bonds.

The solvent DMF (N,N-Dimethylformamide) was purchased from Guangzhou Reagent Company (Guangzhou, China).

The pDNA (ANG-P2A-GFP, angiopoietin-2A peptide-greenfluorescent protein) was purchased from Vigene Co. (Beijing, China). TMC (Chitosan with a 91% degree of deacetylation) was donated by BeiAo Co. (Guangzhou, China). The cell culture medium was RPMI 1640 (Gibco, Grand Island, NY, USA), which contains a 1% penicillin–streptomycin solution (Gibco, USA) and 10% fetal bovine serum (FBS, Gibco, USA), and the culture conditions were 37 °C in a 5% CO_2_ atmosphere.

### 2.2. Preparation of pDNA Loaded Fibers

The PU/LC/pDNA composite was prepared according to the following processing. Briefly, 1 mg of TMC (N,N,N-trimethyl chitosan) and 2 mg of pDNA were mixed and dissolved into the DMF solution. After 1 mL phosphate buffer (PBS, Gibco) contained vertebrae for one minute, the TMC/pDNA mixture was emulsified in the PU/LC solution (15 mL, DMF, PU 5 wt%). After the fiber mats were electrospun by the emulsion, the mats were placed into a vacuum drier for 24 h to obtain completely dry samples (voltage 20 kV, distance 40 cm and room temperature). The plasmid-loaded fibers with nanoparticles were prepared and named PU/LC5%/pDNA, PU/LC10%/pDNA and PU/LC20%/pDNA, respectively. No pDNA-loaded fibers were prepared as a control group.

### 2.3. Morphology of the pDNA Polymer Fibers

The microscopic morphology of the electrospun microfibers was directly observed under SEM (scanning electron microscopy, Tokyo, Japan). All the samples were deposited on suitable holders, and sputtered using gold–palladium under a vacuum before observation. All the samples were carried out at a voltage of 14 kV.

The compatibility of the liquid crystals and polyurethane and the dispersion of the liquid crystals in polyurethane were examined by POM (polarized optical microscopy, Olympus, Tokyo, Japan) and SEM. After the electrospun fibers were produced, the fibers were cut into small pieces, and directly viewed under an optical microscope.

### 2.4. Characteristics of pDNA Release in Electrospun Fibers

We cut the fiber plate containing pDNA into small square pieces (1 × 1 cm^2^), with a mass of about 100 mg. Then, the square of the mats was immersed in 3.0 mL of PBS with a pH of about 7.4. The suspensions were stored in a shaking water bath container maintained at 37 °C. The amount of pDNA released from the fibers was detected by Hoechst 33,258 dye.

### 2.5. Growth of Cells on pDNA-Loaded Fibers

After punching a disc with a diameter of 25 mm, the disc was fixed on a glass with a cell culture ring with an inner diameter of 22 mm. The mats were sterilized by exposure to ultraviolet light. Before adding the medium, the cell seed pad was cultured at 37 °C for 4 h to make the cells adhere to the fibers. A fresh medium was supplemented every two days.

### 2.6. Efficiency of Fiberboard Cells

The expression of GFP (Green Fluorescent Protein) in the HUVEC was observed by a fluorescence microscope (Leica DMR HCS, Weill, Frankfurt, Germany). After cell growth was complete, we collected the mats at a predetermined time. Then, the mats were observed with a fluorescence microscope. The fibers of the cells infected by the plasmid were washed three times by PBS and homogenized in a lysis buffer (Altanta, GA, USA), and carefully collected and preserved in ice.

## 3. Results and Comments

### 3.1. Fiber Morphology

The condition under which all the fibers were prepared was room temperature, and the LC were in the liquid crystal phase. The obtained fibers were transparent and isotropic so that the birefringent liquid crystals could be directly observed by a polarizing optical microscope (POM), as shown in Figure 1. The composite fibers were thermally stable enough to allow repeated heating to the isotropic phase of the liquid crystals contained in the fibers, and the appearance of the fibers did not change after cooling to room temperature (liquid crystal phase) again. As shown in Figure 1, after heating the liquid crystals to the isotropic phase, the white fibers under POM turned dark. After cooling to room temperature, there was no significant change in the macroscopic appearance.

### 3.2. Influence of Applied Voltage

The study of the pattern of the fibers showed that the annular flow with a smooth interface appeared only when the electric field intensity was higher than a certain threshold. It could also be seen from some other studies that the fiber diameter was not coupled with any substantial decrease in the fiber diameter when the field strength increased.

Examples of our electrospun fibers can be observed more intuitively in Figure 2 and Figure 3 (LC 10 wt%). Figure 2 showed the fibers at different magnifications. Figure 3 showed the fibers with different densities.

In this research, fibers were relatively uniform, at a voltage of around 14 kV. When the voltage became higher, the diameter of the fibers decreased. With the increase in the electric field intensity, the jet of the polymer electrospun liquid had a larger surface charge density, and therefore had a greater electrostatic repulsion. Meanwhile, the higher the electric field intensity, the greater the acceleration of the jet. All of these factors could cause the jet and the fiber to have greater tensile stress, leading to a higher tensile strain rate, which was conducive to the preparation of finer fibers. Figure 4 showed the dependence of the average fiber diameter of these samples, measured by an optical microscope at the applied voltage. It could be clearly seen from the figure that, as expected, the diameter decreased with the increase in field strength. Our results revealed that the effect of the change on the diameter of the fibers was negligible, which could explain why we did not find any effect on the phase transition temperature. The phase transition temperature was the same for all the samples shown, and its trend was consistent with many previous studies.

Since our observation of the fibers was only indirect and must be confirmed by direct measurement, the results could not be completely obtained as compared to the composite fiber of TiO_2_/PVP, which uses mineral oil as the core. In the future study, we wish to modify our processing procedures and/or material combinations to study the fibers with respect to the LC core diameter by electron microscopy.

Figure 4 showed the relationship between the fibers’ diameter and the applied voltage (LC 10 wt%). The conductivity of the solutions was around 0.1 Ms/cm, as measured by a conductivity meter (SIN-TDS210, Hangzhou, China). The addition of liquid crystals had little effect on the conductivity of the solution. As the applied voltage increased, the diameter of fibers became smaller. The electric field intensity and the charge density of the solution increased as the applied voltage increased, so the repulsion force was larger and the same as the acceleration with the higher electric field intensity. The increases in repulsion force and acceleration were conducive to the formation of finer fibers. This result was consistent with the results of many previous studies on polymer complex fibers.

### 3.3. Influence of Flow Rate

The diameter of the fibers showed little relation to the solution’s velocity, while the homogeneity decreased with the increase in the solution flow rate, as shown in Figure 5 (LC 10 wt%). In the process of electrospinning, the speed of the spinning was determined by the speed of the syringe. The speed of the adjustments to the spinning speed not only determined the production efficiency of the fibers, but more importantly, also affected the stability of the droplets and the diameter of the fibers. In the process of preparing the fibers, the diameter of the fibers gradually increased with the increase of spinning speed, and even led to the formation of bead-like fibers (as the fiber jet did not completely dry during the flight between the tip and the metal collector).

After adding the pDNA into the LC/polymer fibers, the morphology of the fibers could be seen in Figure 6 (LC 10 wt%). Due to the very small amount of pDNA, it had almost no effect on the morphology and the dispersion of the liquid crystals in PU. The morphology and the dispersion of the liquid crystals were almost the same as that of before the addition of pDNA. The POM images with a compensator could better display the orientation of liquid crystal, and the OM images showed that the fibers owned good transparency after adding pDNA.

### 3.4. SEM Micrographs

In Figure 7, the SEM micrographs showed the microscopic morphology of the non-woven fiber mats. The surface of the PU, PU/LC10% and PU/LC10%/pDNA could be observed clearly. The morphology of the electrospun fibers was similar to that of other studies, such as PLA fibers [44,45,46], PCL fibers [47,48] and PHB fibers [49,50]. The morphology of the fibers appeared to be unaffected by the LC and pDNA, as the appearance of all the samples was nearly the same. It was worth noting that the appearance of the individual fibers was consistent along the length of the fibers. In addition, the diameter of the fibers did not change significantly after adding LC and pDNA. There was uniformity on the fibers according to the SEM micrographs, which could be attributed to the small ratio between the needle diameters and the spinneret. In other reports, however, other needle diameter ratios (0.35–0.64) were used. We thought that the smaller spinneret needle ratio used here allowed for better consistency, as the smaller diameter required a higher speed to obtain a consistent discharge, which resulted in more uniform fibers. Therefore, the results here revealed that a small proportion of spinneret needles could be used as a means to optimize filling in the electrospinning method. Thicker fibers with diameters ranging from several hundred nanometers to one micron could be obtained using these electrospinning parameters. When hydrophilic substrates such as those used here were used, the cylindricity deviation of the PU was seen to occur at high flow rates. It has been suggested that the electrospun PU was closer to a gelatin state just after the initial impact with a surface, and was soft enough to adhere to and spread out along a hydrophilic surface via capillary force. This diffusion gave the electrospun fibers a more elliptical appearance.

### 3.5. Contact Angle and Mechanical Properties of Composite Fibers

Figure 8 showed the water contact angle, tensile stress (MPA), fracture energy, elongation at break of polyurethane electrospun mat and PU/LC electrospun mat with different LC content (%). The increase in liquid crystal content led to the mat being more hydrophilic. Compared with pure PU, the tensile yield stress of PU-containing liquid crystal decreased. The higher the content of the liquid crystal, the more obvious the decrease of the modulus. The tensile modulus decreased from about 0.06 MPa of pure PU to about 0.035 MPa, with a decrease of about 42%. Compared with pure PU, the breaking energy of pure PU increased from 15,236 J/m^2^ to 19,582 J/m^2^, and the elongation at break increased from 523% to 712%.

### 3.6. In Vitro Emission Characteristics of the pDNA-Loaded Fibers

Figure 9 showed the relationship between the pDNA encapsulation and the in-vitro culture time. The pDNA released from the fibers showed a continuous stable period for up to 10 weeks. More than five percent of the pDNA was released within the first four days in all the samples. Over the next three weeks, fifteen percent was released in total. Previous studies have reported that the pDNA was encapsulated with poly (D,L-lactic glycolic acid) (PLGA) fibers by suspension, and the pDNA was suddenly released in 2 h [51]. Moks encapsulated the pDNA into PLGA nanospheres [52]. PEG was used to support the solubility of the pDNA in organic solvents. A total of 46.3% of the first outbreaks were found after 1 h incubation.

In this study, the release of the pDNA core was considered rather slow, compared to layered fibers and suspended electrospun fibers and spheres. As shown in Figure 9, LC was added to the polymer matrix. The initial concentration of the pDNA solution was low and the same as the subsequent accumulation, proving the important role of liquid crystals in the fibers. Compared to the LC5% fibers, the LC10% and LC20% fibers showed a slower drug-controlled release.

Different properties of polymer-based fibers containing pDNA have been studied with different ratios. The first version was the pDNA polymer on or near the glass fibers. The pDNA was released quickly after immersion in an aqueous solution [53]. The pDNA polymeride incubated here was hydrophobic. After being dissolved in the fiber matrix instead of naked pDNA, even the matrix showed no obvious diffusion to the distribution medium. The polymer was removed in large quantities. In addition, there were not only interactions between the matrix polymer PU and pDNA, but the liquid crystals also delayed the release of the pDNA complex.


*The release rate of the pDNA on the pure PU fibers and PU/LC*
*microfibers was uniform in both cases. In addition, the cumulative release amount reached about 40% after 70 days for the samples containing LC, which was observably lower than that of the PU microfibers. The fibers and LC could prolong the pDNA activity, and were beneficial in potential application in medical fields.*


### 3.7. Transfection Efficiency of Cells on pDNA-Loaded Fibrous Mats

The difference in the transfection efficiency of the pDNA-loaded fibers was assessed by the expression of pANG (angiogenic promoting) in HUVEC, since the fluorescence emitted from the proteins could be detected by CLSM. As shown in Figure 10 and Figure 11, the expression of the angiopoietin-2A peptide-greenfluorescent protein increased after 3, 5 and 7 days of culture on the fibers. This finding indicated that fibers owned better transfection efficiency than solid mats without porous structures. Furthermore, after 7 days of culture, the cell transfection efficiency on the PU/LC fibers loaded with pDNA particles was significantly higher than that of cells directly on the TMC/pDNA complex as a positive control (CON). The protein expression on the fibers was higher than that of the TMC/pDNA complex after 3, 5 or 7 days of incubation. This result indicated that the precipitated pDNA polyplexes in the endothelial cell medium lost their activity, resulting in a decrease in transfection efficiency. In contrast, the pDNA released periodically from the fibers remained structurally intact, as well as the transfection ability, which resulted in a continuous increase in the target protein expression. In addition, the transfected nanoparticles could not be bonded to the host genome, so transfection diminished as the cells proliferated.

In this study, the pDNA released could be controlled in both pure PU fibers and PU/LC fibers. The released pDNA from the fibers could transfect cells, and continuously stimulate the secretion of proteins. The pDNA/TMC complex was loaded into PU/LC porous fibers rather than solid films, which showed a more positive effect on the transfection efficiency. In addition, the fibers with a porous structure not only significantly enhanced the surface area, but also maintained the activity of the pDNA, which contributed to maintaining effective local concentrations and a continuous release.

### 3.8. Cell Growth on the Fibers

According to our results, LC-modified PU fibers were used to encapsulate the pDNA, which could enhance the proliferation and adherence of the HUVEC on the fibers. In order to study the response of the HUVEC on pDNA-loaded films, we cultured HUVEC on fibers with different compositions and different liquid crystal contents. Both LC and pDNA played important roles in this system, due to the cell proliferation rate being faster than that of the blank control group.

An optical microscope was used to observe the morphology of the HUVEC directly. The adhesion and diffusion of the HUVEC were shown in Figure 12 and Figure 13. The proliferation of HUVEC at 1, 3 and 5 days was detected by the CCK-8 method; CLSM and ZEN software (Carl Zeiss, Jena, Germany) were used to detect the expression of vinculin, which was shown in Figure 14. 

It was clearly observed that the cells cultured on different samples showed different proliferation and spreading behaviors. The HUVEC cultured on the PU/LC fibers showed better adhesion and diffusion properties than those cultured on the pure PU fibers. Moreover, as the content of LC increased, the cells showed a faster proliferation rate, indicating the important role of LC in the composite fibers, which enhanced the hydrophilicity of the composites. The recombination formed by pDNA was the start of the signaling events that could lead to cell growth, and ultimately to tissue healing. The expression of the pDNA in the fibers of the LC groups with a slower release rate was significantly different from that of the control group, resulting in the aggregation of the increased endothelial cells, as shown in Figure 12 and Figure 13. In conclusion, the drug-loaded fibers showed a positive effect on cell growth, as the pDNA could be slowly released, which could stimulate the cell secretion of vinculin. In addition, the cells in all groups showed high-density proliferation after 3 to 5 days of incubation. Surprisingly, the HUVEC proliferated around these spaces, forming and maintaining enclosed spaces on day 6. As pDNA played a crucial role in vascular development and angiogenesis, it could be inferred that these enclosed spaces led to vascularization and endothelialization.

In conclusion, the composite fibers produced in this study embedded with pDNA could promote HUVEC proliferation, significant vinculin expression and pseudofoot elongation, thus accelerating endothelialization.

## 4. Conclusions

In this study, PU/LC fibers loaded with pDNA were prepared by electrostatic spinning. The addition of liquid crystals and pDNA did not affect the microstructure of the fibers, indicating good compatibility of LC and PU. Using this delivery system, the cumulative release of pDNA from the PU/LC mat sustainably reached around 50%, and the release time was nearly 70 days. The inhibition of cell proliferation and adhesion showed that the PU/LC mat could better preserve the biological activity of pDNA than the pure PU mat. The rabbit model (Appendix A) reveals that LC owns good biocomtibility. Moreover, the anti-adhesion effect of the electrospun mats as a barrier has been identified as a clinical concern. The results showed that the PU/LC mat can protect the biological activity of pDNA, and is a promising platform for inhibiting pDNA deposition and downstream pDNA expression.

## Figures and Tables

**Figure 1 pharmaceutics-14-02489-f001:**
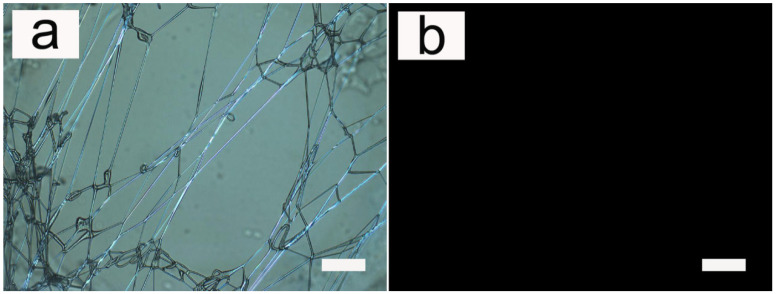
Cross-polarized optical microscopy photographs of PU fibers containing liquid crystals at different temperatures. (**a**) Fibers under room temperature with LC (10 wt%) in their liquid crystal phase; and (**b**) fibers under high temperature with LC in their amorphous phase. The scale bar represents 100 μm.

**Figure 2 pharmaceutics-14-02489-f002:**
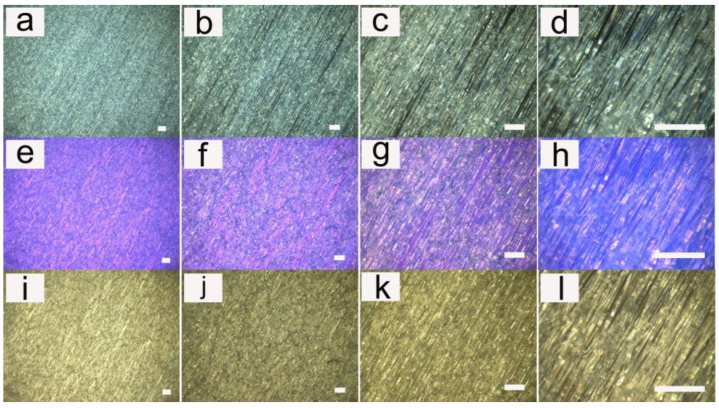
Cross-polarized optical microscopy photographs of PU fibers containing liquid crystals with different magnifications. (**a**) Magnification of 5×; (**b**) magnification of 10×; (**c**) magnification of 20×; (**d**) magnification of 50×; (**e**–**h**) are the corresponding images with a compensator; and (**i**–**l**) are the corresponding optical images without a polarizer plate. The scale bar represents 100 μm.

**Figure 3 pharmaceutics-14-02489-f003:**
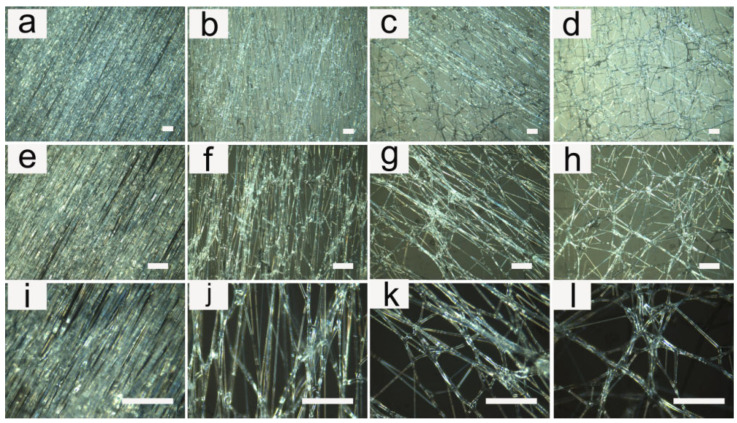
Cross-polarized optical microscopy photographs of PU fibers containing liquid crystals with different fiber densities and different magnifications. (**a**–**d**) Magnification of 10×; (**e**–**h**) magnification of 20×; and (**i**–**l**) magnification of 50×. The scale bar represents 100 μm.

**Figure 4 pharmaceutics-14-02489-f004:**
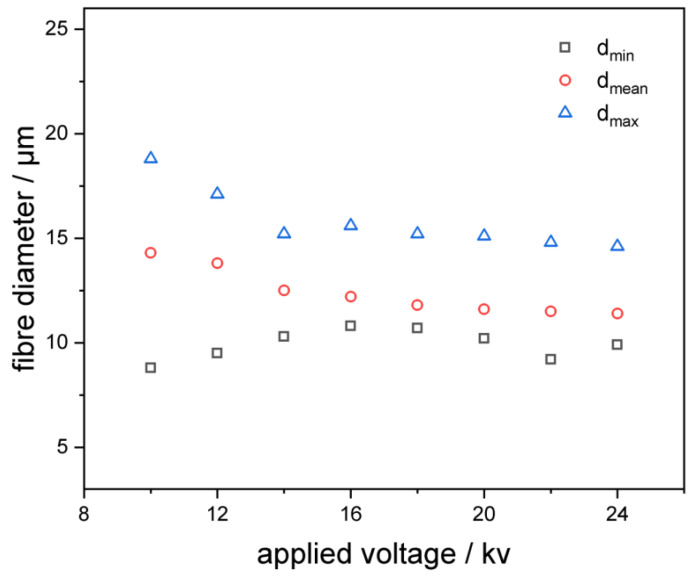
Relationship between the fibers’ diameter and the applied voltage (each sample was measured for 50 times; the standard deviation was 2.8, 2.3, 1.6, 1.7, 1.3, 1.2, 0.9, 1.1).

**Figure 5 pharmaceutics-14-02489-f005:**
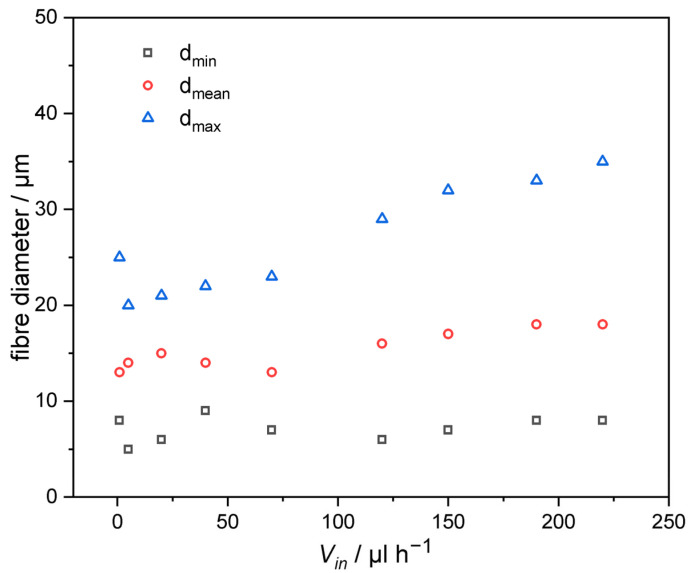
Relationship between the fibers’ diameter and the flow rate of the solvent (each sample was measured for 50 times; the standard deviation was 3.1, 2.9, 2.9, 2.1, 2.2, 2.8, 3.3, 3.5).

**Figure 6 pharmaceutics-14-02489-f006:**
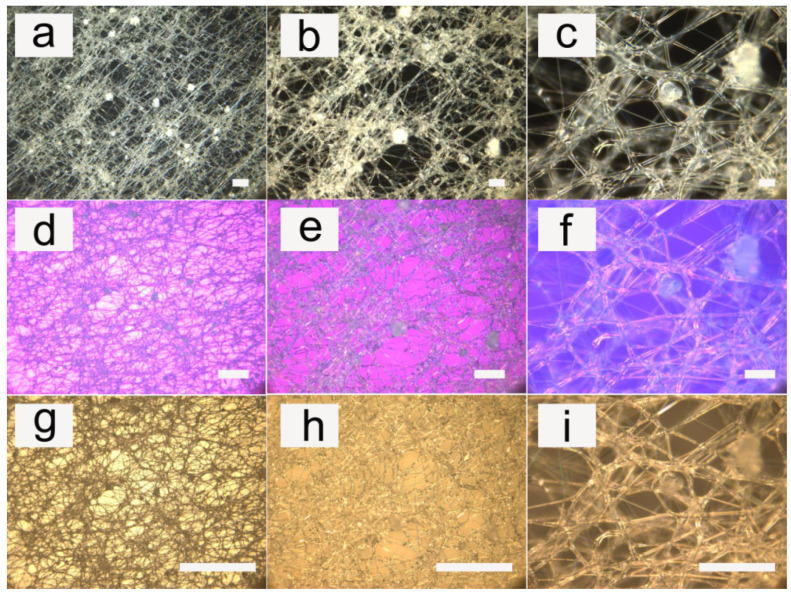
Cross-polarized optical microscopy photographs of PU fibers containing liquid crystals with the addition of pDNA at different magnifications. (**a**) Magnification of 10×; (**b**) magnification of 20×; (**c**) magnification of 50×; (**d**–**f**) are the corresponding images with a compensator; and (**g**–**i**) are the corresponding optical images without a polarizer plate. The scale bar represents 100 μm.

**Figure 7 pharmaceutics-14-02489-f007:**
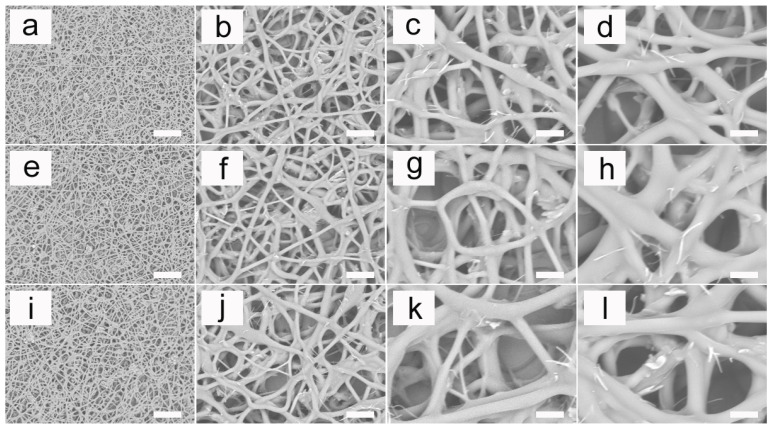
SEM micrographs of the electrospun fibers. (**a**–**d**) are different magnifications of pure PU fibers; (**e**–**h**) are different magnifications of PU/LC10% fibers; and (**i**–**l**) are different magnifications of PU/LC20%/pDNA fibers. The scale bar represents 100 μm.

**Figure 8 pharmaceutics-14-02489-f008:**
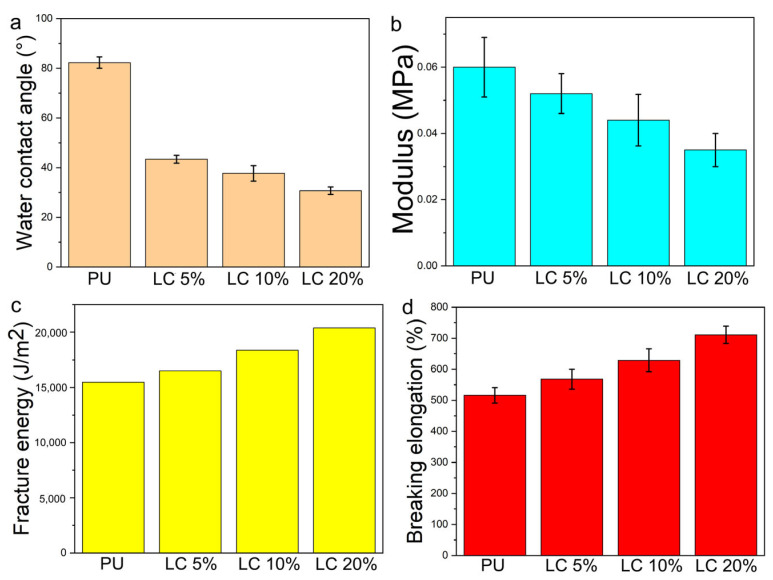
Contact angle, elastic modulus, breaking energy and elongation at break of electrospun mats, (**a**) water contact angle; (**b**) modulus; (**c**) fracture energy; (**d**) breaking elongation.

**Figure 9 pharmaceutics-14-02489-f009:**
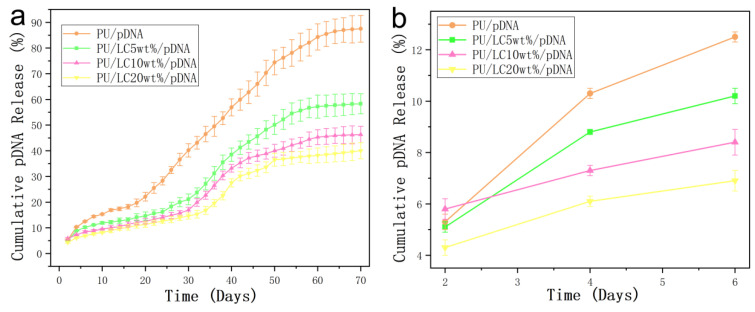
The cumulative pDNA release curve from the complex fibers: (**a**) a long period of release up to 70 days; and (**b**) a short period of release of 6 days.

**Figure 10 pharmaceutics-14-02489-f010:**
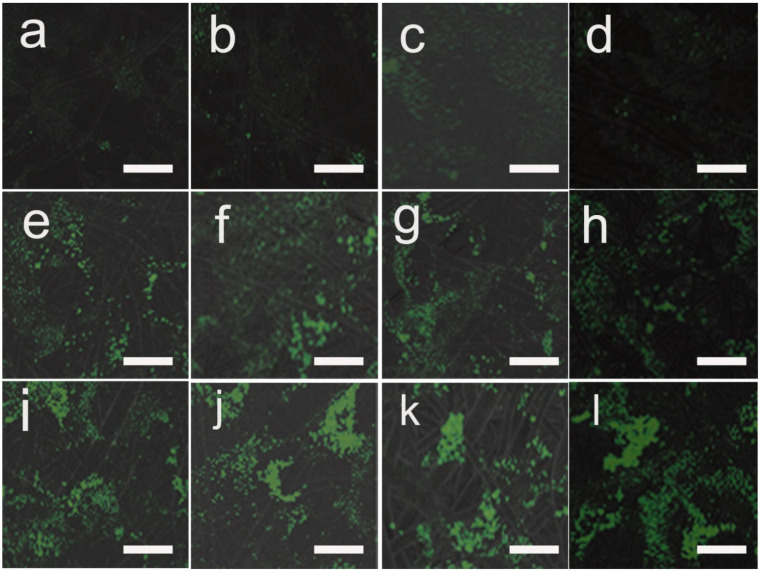
Comparison of the transfection efficiency of microfibers and green fluorescent proteins. Green fluorescence intensity of the pANG fusion protein of the (**a**,**e**,**i**) PU fibers; (**b**,**f**,**j**) PU/LC5% fibers; (**c**,**g**,**k**) PU/LC10% fibers; and (**d**,**h**,**l**) PU/LC20% fibers at 3, 5 and 7 days. The green fluorescence indicates positive transfection. The scale bar represents 50 μm.

**Figure 11 pharmaceutics-14-02489-f011:**
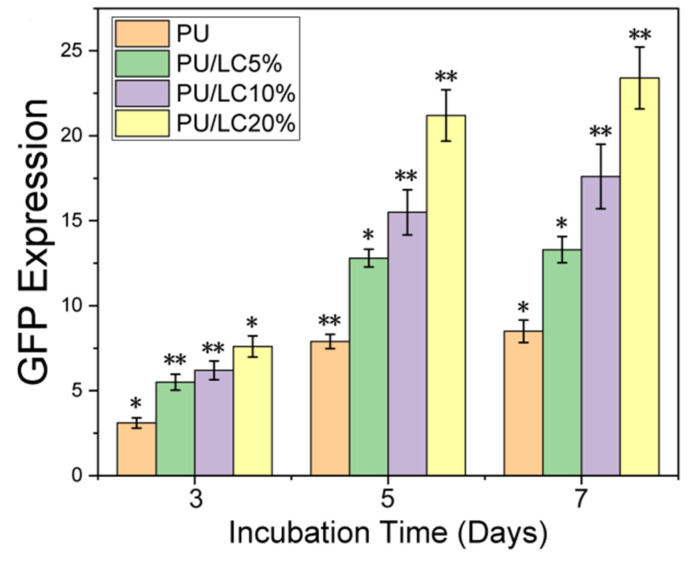
Green fluorescence intensity of pANG fusion proteins at 3, 5 and 7 days. * *p* < 0.05 and ** *p* < 0.01.

**Figure 12 pharmaceutics-14-02489-f012:**
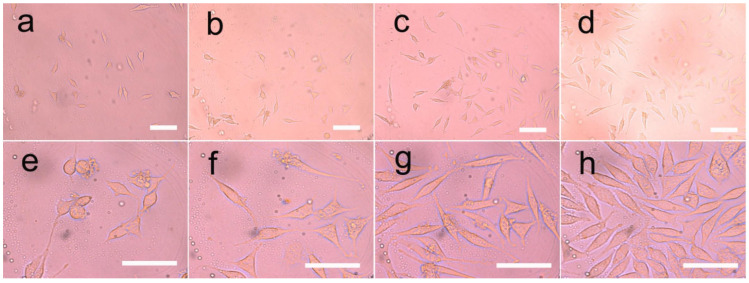
Growth of HUVEC in three-dimensional liquid crystalline polyurethane films. The cells were grown on (**a**,**e**) pure polyurethane; (**b**,**f**) polyurethane with 5% liquid crystal; (**c**,**g**) polyurethane with 10% liquid crystal; and (**d**,**h**) polyurethane electrospun films with 20% liquid crystal. Cell culture was 3 days and 5 days. The scale bar represents 100 μm.

**Figure 13 pharmaceutics-14-02489-f013:**
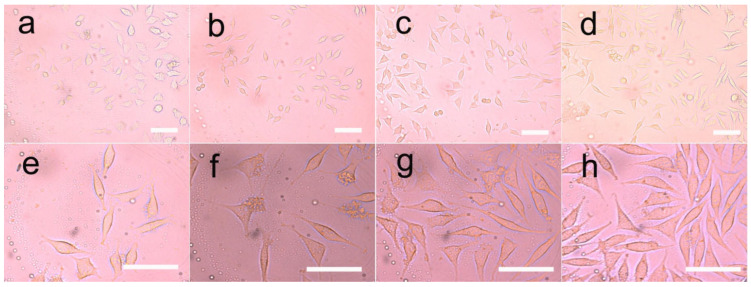
Growth of HUVEC in three-dimensional polyurethane films containing plasmids and liquid crystals. The cells were seeded on (**a**,**e**) pure polyurethane; (**b**,**f**) polyurethane with 5% liquid crystals; (**c**,**g**) polyurethane with 10% liquid crystals; and (**d**,**h**) polyurethane electrospinning films with 20% liquid crystals with a plasmid content of 2 PPM. Cell culture was 3 days and 5 days. The scale bar represents 100 μm.

**Figure 14 pharmaceutics-14-02489-f014:**
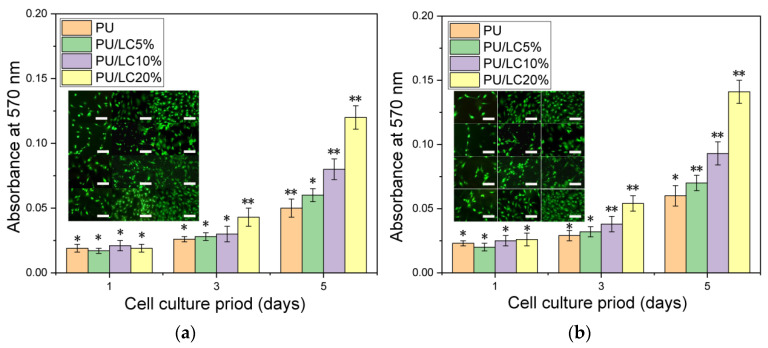
The proliferation of HUVEC cultured on (**a**) pure PU, PU/5 LC %, PU/LC 10% and PU/LC 20% films; and (**b**) corresponding films with the addition of pDNA. * *p* < 0.05 and ** *p* < 0.01. The scale bar represents 100 μm.

## Data Availability

The data presented in this study are available on request from the corresponding author.

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
