# Peer review of "Polyurethane/Liquid Crystal Microfibers with pDNA Polyplex Loadings for the Optimal Release and Promotion of HUVEC Proliferation"

_pharmaceutics, 2022, doi:10.3390/pharmaceutics14112489_

Round 1

Reviewer 1 Report (Previous Reviewer 3)

I accept Liquid crystal elastomer in the reference is different from our system, as stated in the cover letter. Then, stress this point in Introduction.

Author Response

Thanks for your professional comments.

It is stated in Introduction.

Reviewer 2 Report (New Reviewer)

Author Response

Thanks very much for your proffesional comments. your comments help a lot.

The abbreviation ECM has been taken into brackets after full name of
extracellular matrix in the text.

"Fiber structures show high potential to promote natural tissue formation not artificial." has been modified.

"There is no one polyurethane but polyurethanes are the group of polymers containing urethane/urea groups. Their chemical structure can differ and depends on kind of monomers taken to synthesis." has been modified.

The abbreviation LCEs has been taken into brackets and put after ‘liquid crystal
elastomers’.

The aim of work has been modified at the end of Introduction.

More details about used PU have been added.

The description of the synthesis has been added.

The description of preparation of pDNA loaded fibers has been modified. The chitosan is used to protect the activity of pDNA and mass ratio of PU in the mixture has been added.

The abbreviation POM has been taken into brackets and put after polarized optical microscopy’.

Section 2.6. has been modified.

The liquid crystal is in amorphous form at high temperature and in dark field under polarized light. And LCs content in Figures 1, 2 and 3. has been added.

Section 3.2, LC content is 10 wt% and it is added in the text for Figures 4 and 5. The standard deviation bars will affect the clarity of the picture, so it's in the form of text.

LCs content has been added in Figures 6.

"The POM images with compensator could better display the orientation of liquid crystal. And the OM images showed that the fibers owned good transparency after adding pDNA." has been added in the text to conclude.

The text "film" has been changed to "mat".

The release profile observed is expected and beneficial in potential application of electrospun material. It has been added in the end of this part.

Description of the Figure 10 has been modified.

This manuscript is a resubmission of an earlier submission. The following is a list of the peer review reports and author responses from that submission.

Round 1

Reviewer 1 Report

Great paper with great potential and interesting to read, however, it needs to be polished before publication.

1-      In the abstract please consider changing this sentence. It doesn’t make any sense. And it is not connected to the previous sentence

” The liquid crystals synthesized show good compatibility with PU (polyurethane).”

2-      -please add Plasmid DNA (pDNA) in the abstract

3-      Please add cell electrospinning method to your introduction as it is the latest novelty in electrospinning and consider adding these references:

·         Nosoudi, Nasim, Christoph Hart, Ian McKnight, Mehdi Esmaeilpour, Taher Ghomian, Amir Zadeh, Regan Raines, and Jaime E. Ramirez Vick. "Differentiation of adipose-derived stem cells to chondrocytes using electrospraying." Scientific reports 11, no. 1 (2021): 1-11.

·         Jayasinghe, S. N. (2013). Cell electrospinning: a novel tool for functionalising fibres, scaffolds and membranes with living cells and other advanced materials for regenerative biology and medicine. Analyst138(8), 2215-2223.

4-      You have just briefly mentioned that “After the fiber mats were electrospun by the emulsion”, please add the details of electrospinning. Including, voltage, distance and temperature

5-      What is 200 L of fibroblasts? I don’t understand the unit of 200L for cells

6-      Do you use radiation to sterilize the fibers? How is that possible without damaging it?

7-      In “figure 4: Relationship between the fibers’ diameter and the applied voltage”, please show the standard deviation and mention the repletion per number.

8-      In Figure 5. please show the standard deviation and mention the repletion per number.

9-      I understand when you claim that The HUVEC cultured on the PU/LC fibers showed better adhesion but how do conclude that they have better diffusion?

10-   Why there is no burst release observed? can you explain?

Author Response

Thank you for your affirmation of the paper and your professional comments.

  1. ” The liquid crystals synthesized show good compatibility with PU (polyurethane).” has been deleted.
  2.  Plasmid DNA (pDNA) has been added in the abstract.
  3. These references have been added.
  4. The voltage, distance and temperature have been added.
  5. Due to our negligence, we are using ultraviolet sterilization method. And it has been modified.
  6. Due to our negligence, we are using ultraviolet sterilization method. And it has been modified.
  7. The standard deviation has been added and the repletion per number has been mentioned.
  8. The standard deviation has been added and the repletion per number has been mentioned.
  9. Cells are more malleable.
  10. According to the results, it is supposed that the interaction between liquid crystal and plasmid is strong.

Reviewer 2 Report

- The approach is interesting and the topic is appropriate for the journal.

-        The work  has a very clear structure and all the sections are well written in a way that is easy to read and understand.

-         However, little modifications and improvements are needed to enhance the quality of the paper.

-        The paper is focused on Polyurethane/Liquid Crystal Microfibers with pDNA Polyplex Loadings for the Optimal Release and Promotion of HUVEC  Proliferation, reporting interesting results.

-        The Introduction and/or discussion section as well as the list of references should be improved according to the above reported comments.

-        With regard to the experimental results, statistical analysis should be performed to better assess some potential differences.

-        The quality of some figures should be improved.

-        The title is adequate and appropriate for the content of the article.

-        The abstract contains information of the article.

-        Figures and captions are essential and clearly reported.

Author Response

Thank you for your affirmation of the paper and your professional comments.

Some references have been added. And some experimental results, statistical analysis have been added.

Reviewer 3 Report

In this manuscript, the authors developed a DNA-loaded matrix based on polyurethane/liquid formulation prepared by electrospinning method. They analyzed the micro structure of the fibers, and the capacity of DNA loading and release.

Although they did experiments vigourously, this study is still immature for published as an original scientific paper. First of all, there are no control data. As mentioned in Introduction, many papers reported the similar structure of microfibers. In this manuscript, it is unclear what is new, or what is the advantage in their system. The authors should clearly claim their originality, and show the comparative data with other systems, or other DNA transfection methods.

The second problem is that the cell data is poor. For example, in Fig. 10, 12 etc, did the cells migrate into the fibers? In the manuscript, the cells were grown on PU or PU/LC films, or grown in three-dimensional polyurethane films, but the photos are difficult to interpretation. For evaluating the cell growth, negative DNA control is also necessary. Fig. 14, the cell proliferation was enhanced with the increase in LC ratios, then what will happen if the LC ratios are further increased? It remains still unclear whether LC 20% might be the optimal conditon.

Author Response

Thank you for your professional comments.

The membrane of pure PU is the control data.

Liquid crystal and polyurethane composite film as a slow-release system is our innovation and advantage.

The cells were cultered on the surface of the fibers. In the future, we will build three-dimensional structures for cells to grow into. Thanks for your advice.

Because the change of the liquid crystal content will influence the mechanical properties of the composite film. We will carry out animal experiments in the future. Too high content of liquid crystal will lead to mechanical properties not up to standard.

Round 2

Reviewer 1 Report

Thank you for addressing my comments.

Reviewer 3 Report

Just by briefly revising the text (including typo…) without any additional data, it is difficult to change the judgement. The authors said “Liquid crystal and polyurethane composite film as a slow-release system is our innovation and advantage. “ but , I guess the use of liquid crystal is not new, e.g. Chem.Eur.J.2018,24,12206–12220, and would like to have more concrete explanation. If the authors plan future experiments, why can’t they add any data now to improve the scientific quality?